

# Repeated response execution and inhibition alter subjective preferences but do not affect automatic approach and avoidance tendencies toward an object

Izumi Matsuda[1] and Hiroshi Nittono[2]

[1] Department of Psychology, Aoyama Gakuin University, Shibuya, Tokyo, Japan
[2] Graduate School of Human Sciences, Osaka University, Suita, Osaka, Japan

## ABSTRACT

**Background:** Repeated action or inaction toward objects changes preferences for those objects. However, it remains unclear whether such training activates approach-avoidance motivation toward the objects, which leads to actual behavior. We conducted a pre-registered online experiment to examine whether approach and avoidance tendencies were affected by the experience of having executed or withheld a button-press response to a stimulus.

**Methods:** Participants ($N = 236$) performed a Go/NoGo task in which they were asked to repeatedly execute a response to a picture of a mug (*i.e.*, Go-primed stimulus) and suppress a response to another picture of a mug (*i.e.*, NoGo-primed stimulus). They then received one of two manikin tasks, which were implicit association tests designed to assess approach–avoidance tendencies. One manikin task measured the reaction times of moving a manikin toward or away from the Go-primed stimulus and the other picture of a mug (*i.e.*, unprimed stimulus). The other manikin task measured the reaction times of moving a manikin toward or away from the NoGo-primed stimulus and the unprimed stimulus. The participants then rated their preference for the Go-primed, NoGo-primed, and unprimed items.

**Results:** The Go-primed item was evaluated as more highly preferable than the unprimed item in the Go condition, while the NoGo-primed item was evaluated as less preferable than the unprimed item in the NoGo condition. In contrast, the mean approach/avoidance reaction times in the manikin task showed no difference between the Go-primed and unprimed stimuli or between the NoGo-primed and unprimed stimuli.

**Conclusion:** When participants repeatedly responded or inhibited their responses to an object, their explicit preference for the object increased or decreased, respectively. However, the effect did not occur in approach-avoidance behaviors toward the object.

Corresponding author
Izumi Matsuda,
izumi@ephs.aoyama.ac.jp

## INTRODUCTION

Action *vs* inaction toward objects changes preferences for those objects (*Chen et al., 2019*; *Schonberg et al., 2014*). This phenomenon has been examined using the Go/NoGo task (for

a recent review, see *Veling et al., 2022*). In the Go/NoGo task, stimuli are presented on a screen, and participants consistently respond to Go stimuli (by pressing a key) and not to NoGo stimuli (by not pressing a key). After the Go/NoGo task, participants tend to evaluate Go items (*i.e.*, stimuli with action in the Go/NoGo task) as more highly preferable and NoGo items (*i.e.*, stimuli with inaction in the Go/NoGo task) as less preferable. This Go/NoGo task has been applied to training to control participants' preferences. Go/NoGo training studies have used stimuli related to food (*Jones et al., 2016*), alcoholic beverages (*Houben et al., 2012*), and smoking (*Scholten et al., 2019*) to control these consumptions.

Although previous studies have shown that repeated action and inaction affect preference or 'liking,' it remains unclear whether these experiences would also affect motivated behavior or 'wanting.' 'Wanting' usually mirrors 'liking,' but these can be dissociated in terms of their neural mechanisms (*Olney et al., 2018*; *Tibboel, De Houwer & Van Bockstaele, 2015*). 'Liking' refers to the hedonic impact of positive experience, with underlying mechanisms that include hedonic hotspots in limbic brain structures that amplify 'liking' reactions; 'wanting' refers to a motivational process that makes reward cues attractive and able to trigger craving for their reward, mediated by larger dopamine-related mesocorticolimbic networks (*Nguyen, Naffziger & Berridge, 2021*). 'Liking' and 'wanting' do not always correspond with each other. For example, in addiction cases, the 'wanting' system becomes hyper-responsive to the stimuli, although the pleasure generated by the stimuli ('liking') tends to decrease (*Robinson et al., 2016*).

Approach–avoidance behavior tasks have been considered as an implicit measure of the 'wanting' system activity (*Tibboel, De Houwer & Van Bockstaele, 2015*). One approach–avoidance task is the manikin task (*De Houwer et al., 2001*; *Krieglmeyer & Deutsch, 2010*; *Krieglmeyer et al., 2010*). In this task, participants are asked to move a manikin toward or away from the stimulus. If the stimulus induces an approach tendency, participants should move the manikin toward it faster than away from it. If the stimulus induces an avoidance tendency, the participants should move the manikin away from it faster than toward it. *Houben et al. (2012)* examined the effect of the Go/NoGo training on alcohol-related stimuli for heavy drinkers using a manikin task, and found no significant effect. However, these stimuli themselves should trigger strong 'wanting' for the heavy-drinker participants, which might consequently mask the effect of the Go/NoGo training on approach/avoidance tendencies. To precisely examine the Go/NoGo task effect on approach–avoidance tendencies, stimuli for the task should have neither strong 'liking' nor 'wanting' values.

In the present study, we investigated whether repeated action and inaction can affect subjective preference (*i.e.*, liking) and automatic approach/avoidance tendencies (*i.e.*, 'wanting'). The participants were given a visual Go/NoGo task in which they were asked to repeatedly execute a response to one object (*i.e.*, Go-primed stimulus) and suppress a response to another object (*i.e.*, NoGo-primed stimulus). They then received the manikin task consisting of the Go-primed stimulus and another new, unprimed object (*i.e.*, Go condition), or the manikin task consisting of the NoGo-primed stimulus and another new, unprimed object (*i.e.*, NoGo condition). They then rated their preferences for each of the three objects presented in the previous tasks.

Based on previous studies, it was predicted that the Go-primed item would be subjectively evaluated as more highly preferable than the unprimed item, while the NoGo item would be subjectively evaluated as less preferable than the unprimed item (*Chen et al., 2019*). Since 'liking' and 'wanting' usually cohere (*Nguyen, Naffziger & Berridge, 2021*), we predicted that the Go-primed item would induce a greater approach tendency than the unprimed item, while the NoGo-primed item would induce a greater avoidance tendency than the unprimed item. However, we also considered the possibility that behavioral approach/avoidance tendencies and subjective preferences would not correspond with each other (*Gable & Dreisbach, 2021*; *Harmon-Jones, 2019*; *Robinson et al., 2016*). For example, repeated inhibition toward a stimulus may induce psychological reactance, which is a motivational state that drives us to restore freedom (*Rosenberg & Siegel, 2018*). If so, repeated inhibition may result in an approach tendency toward the stimulus.

## MATERIALS AND METHODS

The following protocols were pre-registered at https://osf.io/f3h8e, where the stimulus materials and obtained data are also available. This study was approved by the Ethics Committee of Aoyama Gakuin University (approval number: AO20-16).

### Participants

A total of 240 participants were recruited through a crowdsourcing company (Lancers, Tokyo, Japan). Informed consent was obtained electronically. Only the persons who read the content of the experiment on the recruitment website and agreed to the terms and conditions were able to participate. They received compensation of 450 Japanese yen for their participation. Half were randomly assigned to the Go condition, and the other half were assigned to the NoGo condition. Our *a priori* power analysis using G\*Power (ver. 3.1.9.2) (*Faul et al., 2007*) revealed that a minimum of 85 participants were required ($d = 0.50$, $\alpha = 0.05$, $1 - \beta = 0.90$) to detect the Go/NoGo training effect for each condition (*Allom, Mullan & Hagger, 2016*). Considering the exclusion rate (23.6%) in an online study (*Camp & Lawrence, 2019*), 120 participants were recruited for each condition.

### Stimuli

Six stimuli were prepared, each of which was a picture of a mug. These stimuli were full-color images with a resolution of $900 \times 850$ pixels, obtained from iStock (https://www.istockphoto.com). Three of the six stimuli were randomly selected and used for each participant's experiment, as shown in Fig. 1.

### Procedure

The experiment was conducted online using Inquisit Web Player 6 (https://www.millisecond.com). As shown in Fig. 1, the participants first received the Go/NoGo task, then the manikin task, and then rated their preference for each item presented in the tasks.

#### Go/NoGo task

Two of the three stimuli were selected for this task. Each of the two stimuli was randomly presented on the display for 1 s, with a jittered inter-stimulus interval of 500 ms (±100 ms)

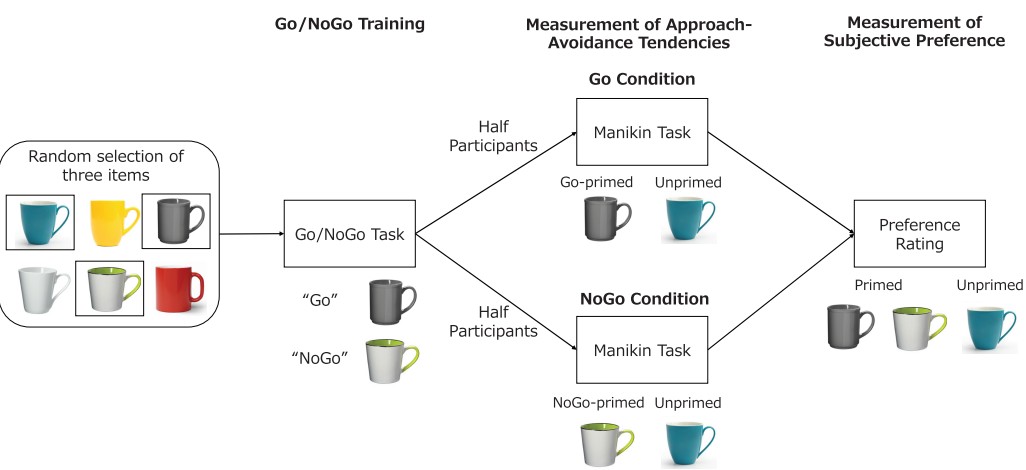

Figure 1 **The flow of the experiment.** Three items were randomly selected for each participant. The participants then received the Go/NoGo task, in which they pressed a key for one of the two items (Go-primed item) and did not press a key for the other item (NoGo-primed item). Half of the participants received the manikin task with the Go-primed item and an unprimed item, and the remaining participants received the manikin task with the NoGo-primed item and an unprimed item. Finally, the participants rated their preference for the Go-primed, NoGo-primed, and unprimed items.

(*Veling et al., 2021*). The participants were instructed to press a response key for one stimulus and not to press the key for the other stimulus. In the practice session, each stimulus was presented four times. This practice session was repeated if the participant pressed the key incorrectly or failed to press the key more than once. In the main session, each stimulus was presented 40 times, which was defined as one block. After each block, a short rest was inserted, during which the display showed the rate of the error trials and an instruction: "We will stop the experiment if the error rate is over a certain threshold." The block was repeated four times. In total, each stimulus was presented 160 times (*Chen et al., 2021*; *Veling et al., 2021*).

### Manikin task

After the Go/NoGo task, the participants received the manikin task, following the protocol used in *Krieglmeyer & Deutsch (2010)*. The manikin task in the Go condition consisted of the Go-primed stimulus for which the participant pressed the key and the last of the three stimuli that was not presented in the Go/NoGo task (*i.e.*, unprimed stimulus).
The manikin task in the NoGo condition consisted of the NoGo-primed stimulus, for which the participant did not press the key, and the last of the three stimuli, the unprimed stimulus.

As described in Fig. 2, each trial began when the participants pressed a key. After 500 ms, a manikin was randomly presented on the upper or lower part of the display. After 750 ms from the appearance of the manikin, one of the two stimuli was randomly presented in the middle of the display. The participants approached the manikin to one stimulus and moved it away from the other using the arrow key. If the participants pressed the arrow key three times, the manikin and stimulus disappeared. The next trial started after 500–1,000 ms. Eight trials (2 stimuli × 4 times) were conducted in the practice

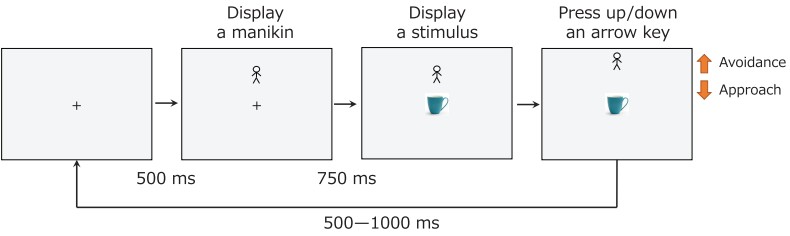

**Figure 2 The flow of a trial in the manikin task.**

session, which was repeated if the participant pressed the wrong key or their response was too slow more than twice. In the main session, 64 trials (2 stimuli × 32 times) were conducted. A rest period was inserted in the middle of the main session. The participants were asked to press the key as quickly and accurately as possible.

After the manikin task was finished, the same task was conducted by changing the combination of the action (*i.e.*, approach/avoidance) and the stimuli. The order of the combination was counterbalanced across participants.

### Preference rating

The participants rated how much they liked each of the three items on a seven-point scale (from 1 = *Don't like at all* to 7 = *Like very much*). The order of the scales for the three items was randomized across participants.

## Exclusion criteria

Participants who met any of the following conditions were excluded from the analysis:

(1) Those who answered that their age was less than 18 years.
(2) Those whose correct response rate was under 85% in the Go/NoGo task (*Johannes, Buijzen & Veling, 2021*). The error trial in the Go/NoGo task was defined as that in which a participant pressed or did not press the key against the instructions.
(3) Those who had more than 25% error trials in the manikin task (*Bertamini et al., 2016*). The error trial in the manikin task was defined as taking more than 1,500 ms to press the first key (*Krieglmeyer & Deutsch, 2010*) or pressing the wrong key (*Renard, de Jong & Pijnenborg, 2017*). For the remaining participants, the error trials were excluded from the analysis.

## Analysis

In the manikin task, the time from the presentation of the manikin to the first key press (*Krieglmeyer & Deutsch, 2010*) and to the third key press (*Bertamini et al., 2016*) were measured. The approach–avoidance index was defined as the difference in the mean reaction time between approach and avoidance trials (*i.e.*, avoidance reaction time–approach reaction time) for each stimulus (*i.e.*, primed and unprimed). A more positive approach–avoidance index indicates a greater approach tendency. A more negative approach–avoidance index indicates a greater avoidance tendency.

The effects of stimulus (primed and unprimed) on the subjective preference score and the approach–avoidance index were examined using $t$-tests in the two conditions (Go and NoGo). Strictly speaking, the seven-point subjective preference scores are ordinal data. However, they can be analyzed using parametric tests (for discussion, see *Norman, 2010*). The effect sizes were described as Cohen's $d$ for $t$-tests. The statistical significance level was set at 0.05. When testing the difference between the two means, the Bayes factor ($BF_{10}$), the odds ratio of the alternative hypothesis (A ≠ B) to the null hypothesis (A = B), was also computed using JASP (https://jasp-stats.org/).

As exploratory analyses, a Condition (Go and NoGo) × Stimulus (primed and unprimed) × Response (approach and avoidance) analysis of variance (ANOVA) was conducted to determine whether the experience of repeated response execution or inhibition affected the reaction times differently in the subsequent manikin task. Also, a Condition (Go and NoGo) × Item (Go-primed, NoGo-primed, and unprimed) ANOVA was applied to the preference scores to determine whether the conditions affected preferences for each item differently.

## RESULTS

After participants who met the exclusion criteria (see Exclusion Criteria section) were excluded, the number of remaining participants was 116 for the Go condition ($M = 43.6$ years, $SD = 9.1$ years, 22–68 years; 73 men and 43 women) and 120 for the NoGo condition ($M = 42.5$ years, $SD = 8.9$ years, 23–68 years; 73 men and 47 women). Table 1 shows the rates of the excluded trials (*i.e.*, error rates) for the remaining participants. Error rates were generally very low (<3%).

### Preference scores

Figure 3 presents the preference scores of each participant and their box plots. In the Go condition, the preference score was significantly higher for the Go-primed item ($M = 4.50$, $SD = 1.33$) than for the unprimed item ($M = 3.99$, $SD = 1.34$), $t(115) = 3.03$, $p = 0.003$, $d = 0.28$, $BF_{10} = 7.80$. There was no significant difference between the NoGo-primed item ($M = 3.73$, $SD = 1.24$) and the unprimed item, $t(115) = 1.50$, $p = 0.135$, $d = 0.14$, $BF_{10} = 0.31$. In the NoGo condition, the preference score was significantly lower for the NoGo-primed item ($M = 4.12$, $SD = 1.32$) than for the unprimed item ($M = 4.53$, $SD = 1.28$), $t(119) = 2.47$, $p = 0.015$, $d = 0.23$, $BF_{10} = 1.85$. There was no significant difference between the Go-primed item ($M = 4.32$, $SD = 1.36$) and the unprimed item, $t(119) = 1.25$, $p = 0.216$, $d = 0.11$, $BF_{10} = 0.22$.

The Condition × Item ANOVA revealed a significant main effect of the condition and the item, $F(1, 234) = 5.60$, $p = 0.019$, $\eta_p^2 = 0.02$, and $F(2, 468) = 8.86$, $p < 0.001$, $\eta_p^2 = 0.04$, respectively. The interaction was also significant, $F(2, 468) = 5.17$, $p = 0.006$, $\eta_p^2 = 0.02$. The *post-hoc* $t$-tests revealed that the preference scores were significantly different between the Go and NoGo conditions for the unprimed item, $t(234) = 3.13$, $p = 0.002$, $d = 0.41$, $BF_{10} = 13.67$, and the NoGo item, $t(234) = 2.30$, $p = 0.022$, $d = 0.30$, $BF_{10} = 1.72$. The preference scores did not differ significantly between the Go and NoGo conditions for the Go-primed item, $t(234) = 1.05$, $p = 0.296$, $d = 0.14$, $BF_{10} = 0.24$.
**Table 1 Error rates in the manikin task.**

| | Error rate (%) | |
| --- | --- | --- |
| | **Approach** | **Avoidance** |
| **Go condition** | | |
| Primed | 2.10 (3.27) | 2.64 (3.21) |
| Unprimed | 1.83 (3.40) | 2.37 (2.96) |
| **NoGo condition** | | |
| Primed | 2.66 (5.67) | 1.90 (3.42) |
| Unprimed | 1.67 (2.43) | 2.68 (5.46) |

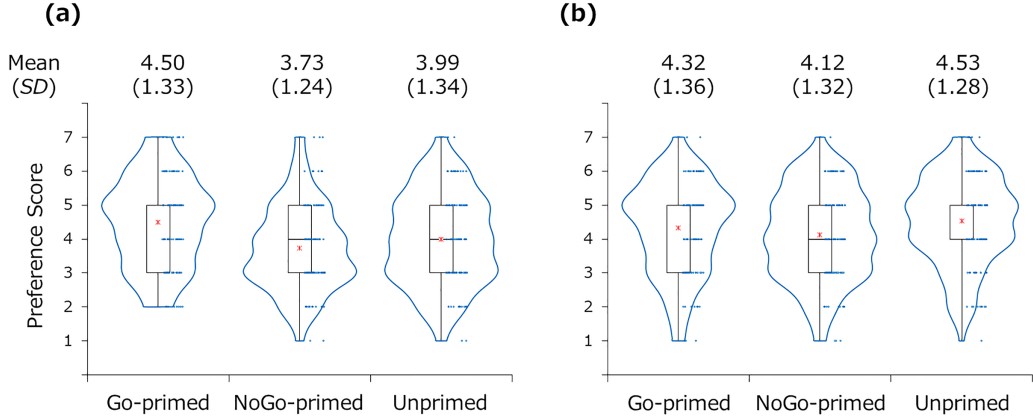

**Figure 3 Preference scores for the Go-primed, NoGo-primed, and unprimed items in the Go and condition (A) and the NoGo condition (B).** The red asterisks indicate the mean.

## Reaction times

Figure 4 presents the reaction times of the first and third key presses for each participant and their box plots in the manikin task. Figure 5 presents the approach–avoidance index (*i.e.*, avoidance reaction time–approach reaction time) for each condition and stimulus. In the Go condition, the mean of the approach–avoidance index was greater for the Go-primed stimulus ($M$ = 53.3 ms, $SD$ = 91.3 for the first key press; $M$ = 53.7 ms, $SD$ = 94.7 for the third key press) than the unprimed stimulus ($M$ = 46.6 ms, $SD$ = 84.1 for the first key press; $M$ = 47.6 ms, $SD$ = 87.1 for the third key press). However, there was no significant difference in the approach–avoidance index between the Go-primed stimulus and the unprimed stimulus, $t(115)$ = 0.51, $p$ = 0.611, $d$ = 0.05, $BF_{10}$ = 0.12 for the first key press; $t(115)$ = 0.45, $p$ = 0.655, $d$ = 0.04, $BF_{10}$ = 0.11 for the third key press. In the NoGo condition, the mean of the approach–avoidance index was smaller for the NoGo-primed stimulus ($M$ = 40.7 ms, $SD$ = 79.2 for the first key press; $M$ = 38.2 ms, $SD$ = 85.3 for the third key press) than for the unprimed stimulus ($M$ = 54.5 ms, $SD$ = 90.4 for the first key press; $M$ = 59.5 ms, $SD$ = 101.7 for the third key press). However, the difference between
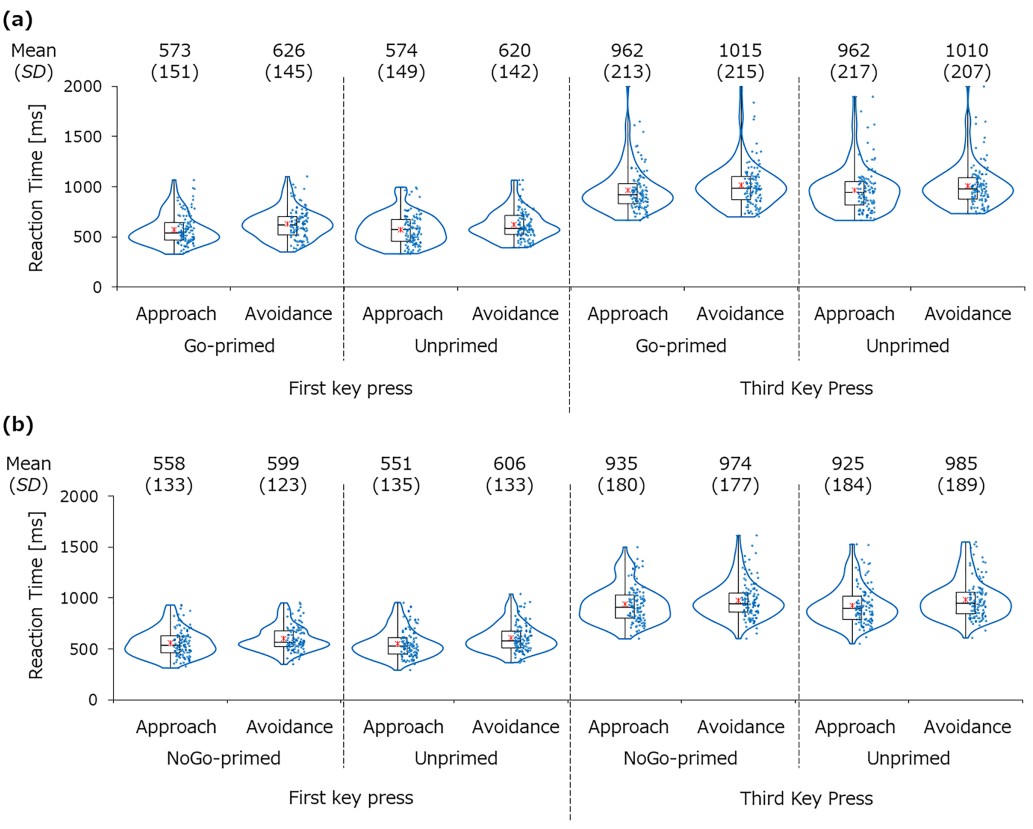

**Figure 4 Reaction times for the primed and unprimed stimuli in the Go condition (A) and the NoGo condition (B).** The red asterisks indicate the mean.

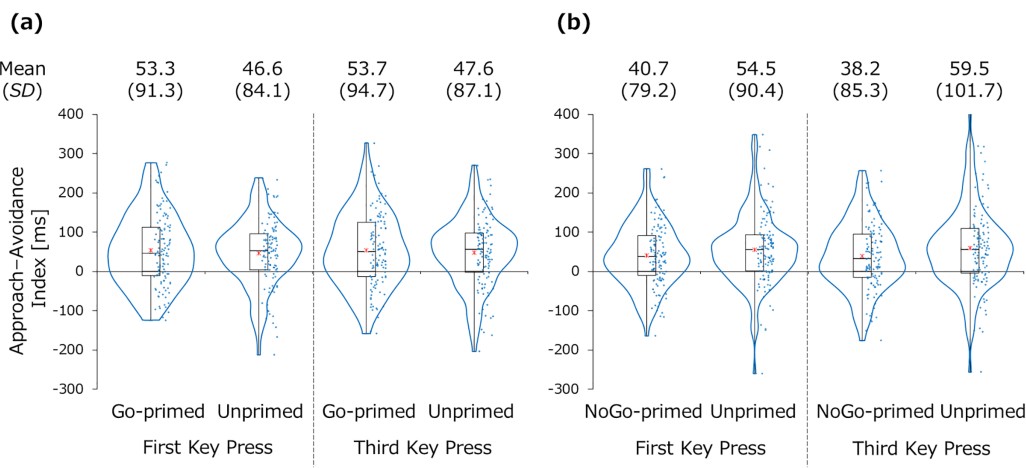

**Figure 5 Approach-avoidance index for the primed and unprimed stimuli in the Go condition (A) and the NoGo condition (B).** The red asterisks indicate the mean.

the NoGo-primed stimulus and the unprimed stimulus was not significant, $t(119) = 1.09$, $p = 0.276$, $d = 0.10$, $BF_{10} = 0.18$ for the first key press; $t(119) = 1.48$, $p = 0.141$, $d = 0.14$, $BF_{10} = 0.29$ for the third key press.

The Condition × Stimulus × Response ANOVA for the third key press did not reveal a significant main effect of the condition or the stimulus, $F(1, 234) = 1.71$, $p = 0.193$, $\eta_p^2 < 0.01$, and $F(1, 234) = 0.11$, $p = 0.746$, $\eta_p^2 < 0.01$. The main effect of Response was significant, $F(1, 234) = 211.90$, $p < 0.001$, $\eta_p^2 = 0.48$. The interactions of Condition × Stimulus, Stimulus × Response, Condition × Response, and Condition × Stimulus × Response were not significant, $F(1, 234) = 0.35$, $p = 0.558$, $\eta_p^2 < 0.01$; $F(1, 234) = 0.59$, $p = 0.443$, $\eta_p^2 < 0.01$; $F(1, 234) = 0.07$, $p = 0.792$, $\eta_p^2 < 0.01$; and $F(1, 234) = 1.91$, $p = 0.168$, $\eta_p^2 < 0.01$, respectively.

## DISCUSSION

The present study investigated subjective preferences and approach/avoidance tendencies when participants repeatedly executed or inhibited an action to a particular stimulus. After the repeated action, the participants evaluated the item as more highly preferable than the unprimed item. After repeated inaction, the item was rated as less preferable than the unprimed item. However, the approach and avoidance reaction times did not differ between the stimulus after repeated execution and the unprimed stimulus. They also did not differ between the stimulus after repeated inhibition and the unprimed stimulus.

As expected, repeated action in response to a specific stimulus made the item preferable to the unprimed stimulus, and repeated inaction made the item less preferable to the unprimed one. A previous study suggested the possibility that repeated speedy responses to the NoGo-stimuli after the training may undo the training effect (*Veling et al., 2022*), but the expected effect of the Go/NoGo training was still observed after the manikin task in the present study. The result of the exploratory analysis shows that the preference scores for the unprimed item and the NoGo-primed item were lower in the Go condition than in the NoGo condition. This might be associated with a previous finding that ignored stimuli are devalued compared to attended stimuli (*Fenske & Raymond, 2006*). In the Go condition, the participants might think of the Go-primed item as a to-be-attended stimulus and the other items as to-be-ignored stimuli, which might lead to undervaluation of the unprimed and NoGo items. In the NoGo condition, both the NoGo-primed and unprimed items were regarded as task relevant and thus to-be-attended stimuli.

In contrast, the reaction times of the approach and avoidance did not differ significantly between the stimulus with repeated action/inaction and the other unprimed stimulus. This result suggests that the Go/NoGo task can change the participants' preferences but does not affect the approach–avoidance behaviors. This is probably because the Go/NoGo responses were nonreinforced and nonincentivized (*Veling et al., 2022*). Even though the value of a stimulus could be updated by performing the Go/NoGo task so as to minimize the gap between the action/inaction decision and the original stimulus value (*Veling et al., 2022*), the update did not involve the prospect of reward. Thus, the mesolimbic reward system was not triggered (*Robinson et al., 2016*) and thus, the 'wanting' process associated with approach–avoidance behaviors was not changed by the Go/NoGo task.

Properties of the stimulus may also have affected reaction times in the present study. We used the pictures of mugs as stimuli, all of which had a handle facing right. Such graspable objects can prime the motor system (*Tipper, Paul & Hayes, 2006*) and cause an

approach tendency toward all items. Although it is known that approach responses are usually faster than avoidance responses in the manikin task (*Krieglmeyer & Deutsch, 2013*; *Paulus & Wentura, 2016*), this bias is usually about 20–30 ms. Figure 5 shows that this bias in the present study was about 40–60 ms. These shorter approach reaction times compared to avoidance reaction times might have masked the Go/NoGo training effect on the approach–avoidance index slightly. However, since this graspable feature was shared across all of the stimuli used in the present study, this would not affect the interpretation of the present results.

## CONCLUSIONS

The repeated action and inaction for a specific item changed the subjective preference but did not induce approach-avoidance behaviors to the object. An object was evaluated highly after repeated action toward it and lowly after repeated inaction when compared to a similar item without action or inaction experience. However, the times for the move toward or away from the primed stimulus did not change significantly with repeated action and inaction. In future research, it is important to be aware of the dissociation between subjective preference reports and behavioral tendencies, which are based on different neural mechanisms of 'liking' and 'wanting'.

### Funding

This study was supported by JSPS KAKENHI grant number 20K03480. The funders had no role in study design, data collection and analysis, decision to publish, or preparation of the manuscript.

### Grant Disclosures

The following grant information was disclosed by the authors:
JSPS KAKENHI: 20K03480.

### Competing Interests

The authors declare that they have no competing interests.

### Author Contributions

- Izumi Matsuda conceived and designed the experiments, performed the experiments, analyzed the data, prepared figures and/or tables, authored or reviewed drafts of the article, and approved the final draft.
- Hiroshi Nittono conceived and designed the experiments, performed the experiments, authored or reviewed drafts of the article, and approved the final draft.

### Human Ethics

The following information was supplied relating to ethical approvals (*i.e.*, approving body and any reference numbers):

This study was approved by the Ethics Committee of Aoyama Gakuin University.

## Data Availability

The pre-registration protocol, stimulus materials, and obtained raw data are available at OSF: Izumi Matsuda and Hiroshi Nittono, Approach and avoidance tendencies toward a particular item after repeated response execution and inhibition. https://doi.org/10.17605/OSF.IO/F3H8E.

## Supplemental Information

Supplemental information for this article can be found online at http://dx.doi.org/10.7717/peerj.16275#supplemental-information.

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
