# Peer review of "Repeated response execution and inhibition alter subjective preferences but do not affect automatic approach and avoidance tendencies toward an object"

_PeerJ, doi:10.7717/peerj.16275_

## Round 0.1 · original submission · Major Revisions

I believe, along with the reviewers, that your article has some potential, and if you can fully respond to the reviewers' comments, we will be happy to consider an appropriately revised version for publication.

I will make a list here to highlight some of the issues based on the comments you will read and my own opinion.

1-It is necessary to emphasize the relevance of the experimental question and the biological value of the study.

2-It is very important to include in the new version a critical view on how the "graspable" nature of the figures might influence the results. In addition, it is important to respond to the consideration that the preference test, always assessed after the performance of the task on the manikin, may have been influenced by the performance of the task itself.

3-It is important to have a figure representing the structure of the manikin task.

4-About terminology: instead of neutral, I would suggest using terms such as primed and not-primed.

5-Please, make more explicit the link to existing literature. In particular, it is unclear which of the sentences are your speculations and which ones are based on previous literature

6-Please, rephrase the hypothesis. Furthermore, the structure of the tested hypothesis seems odd:
H1 and H2 are very generic and H2 presents two opposite potential results (b and c). So, once one of the two is observed, what it the change in the theoretical construct?

7- Likert scale: please add a non-parametric test for the evaluation. I know that most studies employ parametric test, but this is an ordinal scale.

Reviewer 1 ·

Basic reporting

In the current manuscript, 236 participants performed a GO-No Task in which they were asked to repeatedly execute an action toward a “neutral” stimulus and to suppress a response to another neutral stimulus. Following this task, participants performed two “manikin” tasks, where they had to move a manikin toward or away the go –primed and the no-go primed stimuli. In addition, they were asked to rate their preference for the go and no go primed stimuli. Despite the fact that I do see some potential in this study, I am afraid that the current version it is not suitable for publications. In particular, the only significant result (a change of preference for an item primed with action or inaction) is not new, and the reaction times results are weak and not convincing for the reasons I will explain below. I had hard times to understand as well the purpose of the study and its biological relevance. Overall the manuscript was difficult to read for me. I found both the introduction and discussion not very fluent, both in terms of concepts and language.
I hope my suggestions will help the authors to reach a more complete version of the study.

Experimental design

- The authors use what they define “Neutral” stimuli. But “neutral" with respect to what? If we were in the context of an emotion study for example, the definition of neutral would be pretty clear (e.g. a neutral expression, in contrast with a happy/negative expression). In this study, which is exploring motor priming/preferences, the chosen stimuli are absolutely not neutral. Without this major consideration the study is conceptually incomplete. On the other hand, I think that in the use of “mugs” as stimuli lies the potential of this study.
Mugs are objects which fall into the category of “graspable” objects. There are solid studies showing that the simple vision of objects that we can grasp can prime the motor system (e.g. Daprati, E., Balestrucci, P. & Nico, D. Do graspable objects always leave a motor signature? A study on memory traces. Exp Brain Res 240, 3193–3206 (2022); Tipper, S.P., Paul, M.A. & Hayes, A.E. Vision-for-action: The effects of object property discrimination and action state on affordance compatibility effects. Psychonomic Bulletin & Review 13, 493–498 (2006), and many more), consequently causing compatibility/incompatibility effects (lengthening/speeding up of reaction times) when the same effector used to manipulate the visible object is also used to give an answer (the hand). Your reaction times results might be influenced by these effects. You contrasted a primed stimulus, a mug, against another new stimulus which is still a mug of a different color. Now, even if the new mug has not been seen but the participants before the manikin task, this can still influence the reaction times as a consequence of the effects described above, and perhaps hide significant effects of your task. I strongly recommend to have a look at the literature I mentioned above, I am sure your study will gain a lot if these concepts are integrated.

- Just a suggestion: if I understood correctly, your upper reaction time to give an answer is 1500 ms. I would try to give less time, this might unravel stronger effects.

- In the description of the tasks, references to figure numbers are missing.

Validity of the findings

- As briefly mentioned, both in the abstract and the introduction, the aim of the study is described as exploring whether repeated action / inaction can induce automatic approach/avoidance behavior towards primed objects, and not simply change the participants’ preference for the primed stimuli as already shown by previous studies. The authors basically replicate previous results, and at the same time do not emphasize enough the negative results of the two manikin tasks.

- The relevance of the experimental question is unclear to me and the biological value of this study is not discussed. Why would the repetition/suppression of an action lead to a preference/approach behavior towards the primed stimulus? What is the physiological advantage of this mechanism? Is it just an epiphenomenon? In my opinion this should be discussed in order to make a useful sense of the findings. Unfortunately, this point is neither explored nor questioned at a deeper level.

Additional comments

- Line 79 to 85: this paragraph should be more clearly explained. In particular, it is unclear which of the sentences are your speculations and which ones are based on previous literature (hence you should insert some references here).
- Line 96. You should argument why repeated inhibition could also increase approach tendencies. If I understood correctly, you are now stating exactly the opposite of what you have described in the previous lines.
- Line 208 - 216 (and through the manuscript): this is perhaps personal taste, but when you cite the different hypotheses, I think it would be easier for the reader to report a full descriptive sentence, rather than “H1A, b “, otherwise one has to go back to the hypotheses description in order to understand what you are talking about.

Reviewer 2 ·

Basic reporting

The authors report the results of a behavioral study in which they investigated the effects of a go/no go training with neutral stimuli on explicit preference and implicit approach/avoidance tendency toward those stimuli.
The paper is well written and structured, although the discussion section could be improved (see part3 Validity of the findings).The raw data are provided and easy to read.
I have two concerns about the figures:
1) The task is clearly explained in the methods section, but I still think it is necessary to add a schematic illustration of the manikin task to Figure 1.
2)all the boxplot figures would benefit from the addition of significance bars between the tested groups, whether the difference is significant or not, to make it easier to understand the results when looking at the figures without always having to resort to text

Experimental design

The experimental design is sound, the design and method employed are well discussed and the experimental hypotheses are explained in a way that is very clear to the reader. The number of experimental subjects is appropriate, and statistical analyses are clearly reported.
The third point in the exclusion criteria section should be clearer: were subjects with more than 25% errors in the manikin task completely excluded, and in the remaining subjects (<25% errors) were all trials with reaction times greater than 1500ms removed? In the latter case, were the errors equally distributed among the experimental conditions?

Validity of the findings

The novelty of the results is difficult to assess. The study reports as its main finding that neutral stimuli are evaluated positively after repeated action and negatively after repeated inaction (thus as an effect of go/no go training). Nevertheless, authors report how the same manikin task could influence preferences toward stimuli(lines 273-276). If preferences were always assessed after the execution of the manikin task, it is not clear to me how the change in preference could be linked to the action/inaction training task, rather than to the execution of the manikin task itself.

A second aspect that needs further clarification concerns the differences with previous studies that found similar results. In the first lines of the introduction (lines 44-50) it is clearly stated that object preferences are influenced by action/inaction toward objects, while the discussion (lines 284-286) reports the opposite result, at least when it comes to food-objects in online-administered experiments. What are the main differences in this study with previous studies? Does the novelty lie in the fact that non-food objects were used or rather that it is online-administered? This is not clear.

The authors report no significant differences in reaction times or apporach/avoidance index that support the hypotheses under study. Looking at figure 3, however, it appears that reaction times during avoidance are always higher regardless of object (primed or neutral) or condition (go or nogo). This is also evident from the anova result that shows response as the only significant factor (line 258-259). This also affects the approach avoidance indexes. Can the authors comment on this? Is there something in the task design or go/nogo training that can account for the systematically greater slowness in avoidance? Perhaps it is possible to repeat the analyses excluding only those subjects who show a significant effect of response in the anova.

The additional analyses seem to show somewhat different results, but they are in my opinion presented in an unclear manner. In the discussion the authors comment on the results of additional analyses that are explained only later in the appendix section. This is confusing for the reader who finds the commentary on the results before the results; the content of the appendix section should be moved directly to the results section, before the discussion. There is also no trace in the conclusion section of the results of the additional analyses, which show at least an indirect effect (correlation between explicit preference and approach avoidance index) and a partial direct effect (only in the nogo condition) when considering reaction times adjusted for error rate. If these results are in any way relevant according to the authors, they should take them into account in the conclusions.

---

## Round 0.2 · Major Revisions

Dear Dr. Matsuda,

I have read the revised version of the article. I agree with you in changing the title, which is now clearer in reporting the main findings.
I still have some comments that I think need to be addressed to improve the communication of your work.

I found the introduction a bit difficult to read. From sentence to sentence new concepts are added that do not seem clearly (explicitly) related to the topic of the paper. Here are some specific comments.
Introduction
Line 48-50: The sentence combines object (food, drink) and behavior (smoking) and ends with a reference to "unhealthy behaviors". It needs to be rewritten, because food and drink are not unhealthy behaviors: some types of consumption are.

Line 51:60, the new added part. I found this part quite confusing, combining diverse elements not in a clear manner. I make a schema of some of the elements below:

1) Action and inaction toward an object update its value and thus the actor preferences.

2) The item(object) associated with a reward promotes action and the item associated with punishment promotes inaction.

Why did the authors introduce the concept of reward and punishment here? Does this imply that the action toward an object is always associated to a reward? This is not clear, and adds a factor (the reward) difficult to integrate

3) When the decisions run counter the tendency promoted by the item this results in prediction error that requires update of the value of the item.

This sentence is too obscure to me. And the concept of prediction error is introduced.

4) The introduction of the concept of de-evaluation and inhibition and the references to frontal circuits are not really clear.

5) Other concepts were added (e.g. incentive salience) but their connection to the action and inaction is not described in a linear, easy to understand manner.

What I have found difficult is mainly the immediate understanding of the connections between the different factors and arguments introduced and the main topic of the article. I can make the connections myself, but this would be too much effort for a reader approaching a scientific journal with a wide audience.
Line 71:72. This is a clearer sentence, and I think the author should work on the previous part to better channel what is needed to get to this point. They could work better on the part about wanting and liking, which seems more related to their work.


Results.
I would present just the analysis planned in the registered report.
In the preference score analysis, I would add the mean and the median (as well SD values). They are in the figure, but you can put them in the text to make everything clear. Furthermore, I would remove from Figure 3 the Go-vs unprimed and Nogo-vs unprimed for both Go and NoGo condition. Of course, the scale of the y axis must change accordingly.
There is no mention in the recorded report of the ANOVA analysis performed in the preference score. Here the authors can decide whether to include this analysis by stating that it is exploratory (and mention the exploratory nature in the topic part of the discussion), or remove it.

Reaction Times
Please add mean and SD
Figure 5, please remove the differences between conditions (Go vs Unprimed, ecc).
In the text must be made explicit that the ANOVA Condition × Stimulus × Response (approach and avoidance) is exploratory, as declared in the registered report.
Discussion
The main result of the paper is reported in the title now.
I think the author can improve the discussion by adding something on the dissociation between the subjective “declared” preference and the avoidance/ approach tendencies.

---

## Round 0.3 · Minor Revisions

Here I will add some remaining comments to finalize your paper.

In relation to the 7 points scale. Please, add some reference justifying the possibility of considering preference as a continuous variable.

Other comments are related to the sentences

Line 52; delete “but”
Line 62 “Approach avoidance “, please “or” between the two.
Line 73: “neither strong liking nor wanting” ..a word is missing at the end of the sentence. Possible candidates can be: value, attribute, feature, …or some other word.
Line 246: better talk about “action”

Line 286. Conclusions 286
I would start the conclusion with the sentence in line 289. Then the sentences in line 286 and 288, and then in 291.
For the sentence now in line 286 something is missing. “ An object was evaluated more highly…”. I would suggest to not use “highly” as the only attribute. .

---

## Round 0.4 · accepted · Accept

The authors have addressed the requests for revision.
I have assessed the revision myself and I am happy with the current version.
The manuscript is ready for publication.

Reviewer 1 ·

Basic reporting

The manuscript significantly improved from the previous version. Concepts are much more clearly explained, and the reading runs way smoother. In this new version, I liked in particular the idea of using the "liking" and "wanting" contrast, which now conveys a clear picture of what the authors want to investigate.

Experimental design

No additional comments

Validity of the findings

No additional comments